# Amp-TB2: An Updated Model for Calcic Amphibole Thermobarometry

Filippo Ridolfi

Institut für Mineralogie, Leibniz Universität Hannover, 30167 Hannover, Germany; filippo.ridolfi@uniurb.it

**Abstract:** Amphibole (Amp) plays a crucial role in the study of several earth and planetary processes. One of its most common applications is in thermobarometry, especially for volcanic-magmatic systems. However, many thermobarometers require the input of melt composition, which is not always available in volcanic products (e.g., partially crystallized melts or devitrified glasses), or show rather high errors for characterizing the depth of magma chambers. In this work, a new version of amphibole thermobarometry based on the selection of recently published high-quality experimental data is reported. It is valid for Mg-rich calcic amphiboles in magmatic equilibrium with calc-alkaline or alkaline melts across a wide P-T range (up to 2200 MPa and 1130 °C) and presents the advantage of being a single-phase model with relatively low errors (P $\pm$12%, T $\pm$22 °C, log$f$O$_2$ $\pm$0.3, H$_2$O in the melt $\pm$14%). A user-friendly spreadsheet (Amp-TB2.xlsx) for calculating the physico-chemical parameters from the composition of natural amphiboles is also reported. It gives warnings whenever the input composition is incorrect or diverges from that of the calibration data and includes diagrams for an easy graphical representation of the results.

**Keywords:** amphibole; thermobarometry; volcanoes; magma chamber; igneous rock; magma feeding system; calc-alkaline magma; alkaline magma

## 1. Introduction

Amphibole (Amp) is one of the most common mafic minerals in the lithosphere, the major component of some types of rocks (e.g., amphibolite and hornblendite), and plays a crucial role in the understanding of several planetary, earth, and health issues [1–7]. The word "amphibole" means ambiguous, from the Greek αμφιβολος (Réné J. Haüy, 1743–1822), and presently refers to a supergroup of complex silicate minerals (more than 100 end-members) with the general formula AB$_2$C$_5$T$_8$O$_{22}$W$_2$ (where A = vacancy, Na, K, Ca, Pb, Li; B = Na, Ca, Mn$^{2+}$, Fe$^{2+}$, Mg, Li; C = Mg, Fe$^{2+}$, Mn$^{2+}$, Al, Fe$^{3+}$, Mn$^{3+}$, Cr$^{3+}$, Ti$^{4+}$, Li; T = Si, Al, Ti$^{4+}$, Be; W = OH, F, Cl, O$^{2-}$) [8]. Amphibole crystal chemistry is controlled by complex mutual relationships between the composition of the system (minerals and melt) and physico-chemical parameters such as pressure (P), temperature (T), water ($f$H$_2$O), and oxygen ($f$O$_2$) fugacity [3,9–11].

Because of the large variety of possible substitution mechanisms, Amp offers many applications in geosciences. For example, the composition of Amp can be used as a tracer for volatile concentrations in melts and fluids, but one of its most common applications is in geothermobarometry, especially for volcanic systems, e.g., [9,12,13]. Recent single-phase thermobarometric developments valid for Mg-rich calcic amphiboles in equilibrium with calc-alkaline or alkaline melts (hereafter single-Amp) have been in growing demand because of their user-friendly application across a wide P-T range [10–13]. In fact, although thermobarometric models based on chemical equilibria among coexisting mineral-mineral or mineral-melt pairs [9,14–17] are useful tools widely used to estimate the P-T path and chemical evolution during igneous processes, their application is difficult whenever different magmas interact within the crust and erupt to form mixed (hybrid) products [18–20]. In addition, the composition of the melt formed at magma storage conditions is not

preserved in many volcanic rocks. These problems favored the development of several single-phase thermobarometric applications [21–24] which, when properly used, better highlight the sub-volcanic processes recorded in the minerals of hybrid products [25,26]. Nevertheless, Ridolfi and Renzulli [10] showed that previous Amp applications indicate large uncertainties when tested against high-quality experimental results obtained in a large P-T range (up to 2200 MPa and 1130 °C), typical of worldwide volcanic plumbing systems.

The single-Amp application, developed by [10], relies on an empirical procedure to estimate the pressure calculated from a series of five barometric equations, obtained through multivariate least-squares regression of literature and experimental data and account for an overall error of about 12%. The same composition of natural Amp is then used to obtain estimates of T ($\pm 24$ °C), $fO_2$ ($\pm 0.4$ log units) and coexisting melt composition by other empirical equations. The applications of the approach proposed by Ridolfi and co-authors [10,13] produce P-T results consistent with independent high-quality experimental data [27–29] and appear to be in agreement with constraints from complementary methodologies, including depth estimates of sub-volcanic magma bodies located by geophysical data [13,19,26,30–37]. Criticisms have been advanced [20,38,39], and replies were given by Gorini et al. together with warnings to avoid amphibole crystallized at disequilibrium conditions and on the correct application of Amp thermobarometry (in their Supplementary discussion, [26]). However, the barometric procedure proposed by [10] works well for low pressures (<500 MPa) and above 900 MPa, but may result in a few shortcomings because of the lack of moderate temperature data at intermediate pressures; only one high-T (1050 °C) amphibole obtained at 700 MPa is included in the calibration dataset of [10].

In this article, an updated version of the single-Amp model of [10], based on recently published high-quality experimental data, is reported together with a user-friendly spreadsheet to calculate crystallizing P, T, $fO_2$, and $H_2O_{melt}$ (volatile content in the melt) conditions of Mg-rich calcic amphiboles in equilibrium with calc-alkaline or alkaline melts.

## 2. Methods

Recent experimental data in literature were first checked according to a series of criteria aimed at selecting high-quality experimental data of amphiboles in equilibrium with the melt [10,13,26,40]. For instance, amphibole and coexisting phases have been checked for their compositional homogeneity by comparing the standard deviations of their multiple analyses (major element oxides measured by Electron MicroProbe (EMP)) to those of experimental amphiboles selected in [10], and the composition of the recently published experimental amphiboles were plotted in a multitude of variation diagrams to check their consistency with the natural magmatic ones [10,13]. In addition, their quality was checked using the application AMFORM.xlsx, allowing the calculation of the Amp formula and reporting of warnings for bad analyses and deviations from the correct stoichiometry [40].

The result of this selection was a series of 12 experimental Amp compositions synthesized by [27–29,41–44] in the ranges of 885–1060 °C, 200–1000 MPa, and 2.4–3.2 ΔNNO ($logfO_2$-$logfO_2$ at Ni-NiO buffer; e.g., [45]). These amphibole data were reported in the electronic supplement Amp-TB2.xlsx together with the other 61 experimental samples selected by [10]. The overall database was used to refine the single-Amp application following the work of [10] that took into account the relative errors of the five pressure equations and minimized the overall barometric uncertainty.

## 3. Results and Discussion

Ridolfi and Renzulli [10] reported five logarithmic and linear P-equations calibrated with Amp data obtained in different pressure ranges (P1a:130–2200 MPa; P1b and P1c: 130–500 MPa; P1d: 400–1500 MPa; P1e: 930–2200 MPa) and defined parameters such as ΔPdb (P1d-P1b) and XPae (P1a-P1e/P1a), a procedure to calculate a P2 (using the results of the above P-equations and parameters), and the final pressure values (i.e., P) based on the

"apparent" percentage error, APE, i.e., (P1a − P2) × 200/(P1a + P2). Since these equations were calibrated within physico-chemical and compositional fields with a homogeneous density of data, new P calibrations (including the newly selected 12 amphiboles) were not needed as they can result in biased equations. However, when the newly selected experimental amphiboles were used to test this procedure, P and T errors were on average 32% and 29 °C and could be as high as 100% and 67 °C, respectively. These uncertainties were much higher than those predicted by the method (see Section 1).

In this new version of single-Amp thermobarometry, the procedure to estimate the final P values of [10] was modified by changing the parameter thresholds as follows:

(i)     if P1b < 335 MPa, then P = P1b;
(ii)    if P1b < 399 MPa, then P = (P1b + P1c)/2;
(iii)   if P1c < 415 MPa, then P = P1c;
(iv)   if P1d < 470 MPa, then P = P1c;
(v)    if XPae > 0.22, then P = (P1c + P1d)/2;
(vi)   if ΔPdb > 350 MPa, then P = P1e;
(vii)  if ΔPdb > 210 MPa, then P = P1d;
(viii) if ΔPdb < 75 MPa, then P = P1c;
(ix)   if XPae > −0.2, then P = (P1b + P1c)/2;
(x)    if XPae > 0.05, then P = (P1c + P1d)/2;
(xi)   if none of the above conditions are satisfied, then P = P1a.

This method indicated an overall pressure uncertainty of ±12% (i.e., $\sigma_{est}$ calculated on the % error of all selected data; Figure 1a), similar to those reported in [10]. APE should be lower than 60% and was only used as a warning in the spreadsheet Amp-TB2.xlsx (see below).

Once P was estimated, T was calculated using the equations two of [10]. The overall T uncertainty of ±22 °C (Figure 1b) results improved slightly with respect to that reported in [10], i.e., ±24 °C (see Section 1).

For $fO_2$ and $H_2O_{melt}$, new equations independent of P and T were calibrated by multivariate least-square analyses [10,24,40] using all the data available in Amp-TB2.xlsx:

$$\Delta NNO = -10.321 \times Al^{IV} + 4.470 \times Al^{VI} + 7.551 \times Ti + 5.463 \times Fe^{3+} - 4.739 \times Mg - 7.203 \times Fe^{2+}$$
$$- 17.561 \times Mn + 13.762 \times Ca + 13.756 \times Na^A + 27.594 \times K \ (R^2 = 0.951) \tag{1}$$

$$\ln(H_2O_{melt}) = -1.375 \times Al^{IV} + 1.710 \times Al^{VI} + 0.859 \times Ti + 1.189 \times Fe^{3+} - 0.676 \times Mg - 0.390 \times Fe^{2+}$$
$$- 6.402 \times Mn + 2.549 \times Ca + 1.371 \times Na^A + 1.257 \times K \ (R^2 = 0.988); \tag{2}$$

where $Al^{IV}$ to K were the atoms per formula unit as calculated using the 13-cations method (see Amp-TB2.xlsx). Equation (1) accounted for an $\sigma_{est}$ of ± 0.3 log units, whereas $H_2O_{melt}$ showed an overall uncertainty of ±14% (Figure 1c,d). Both uncertainties were lower than those reported by [10] who used a smaller amount of data.

All of the above calculations can be easily performed using the spreadsheet Amp-TB2.xlsx (Supplementary Materials), in addition, reporting warnings in case the stoichiometry of the input amphiboles is not satisfied (most probably wrong analyses) and/or the composition diverges too much from those of the experimental amphiboles selected in this work. In these cases, the uncertainty of the calculated physico-chemical parameters is unknown, and the input amphibole data should be discarded or treated with caution as very strong deviations can occur when any amphibole composition departs from the experimental database used for calibrating an empirical thermobarometer (e.g., [10]). For this reason, Amp-TB2.xlsx reports warnings whenever the input Amp is outside the compositional ranges used to calibrate the model. This is particularly useful to evaluate the quality of the output ΔNNO and $H_2O_{melt}$ values since Equations (1) and (2) are calibrated using lower amounts of data (25 and 48, respectively; Figure 1c,d) representing smaller compositional ranges; see Amp-TB2.xlsx for practical and additional information.

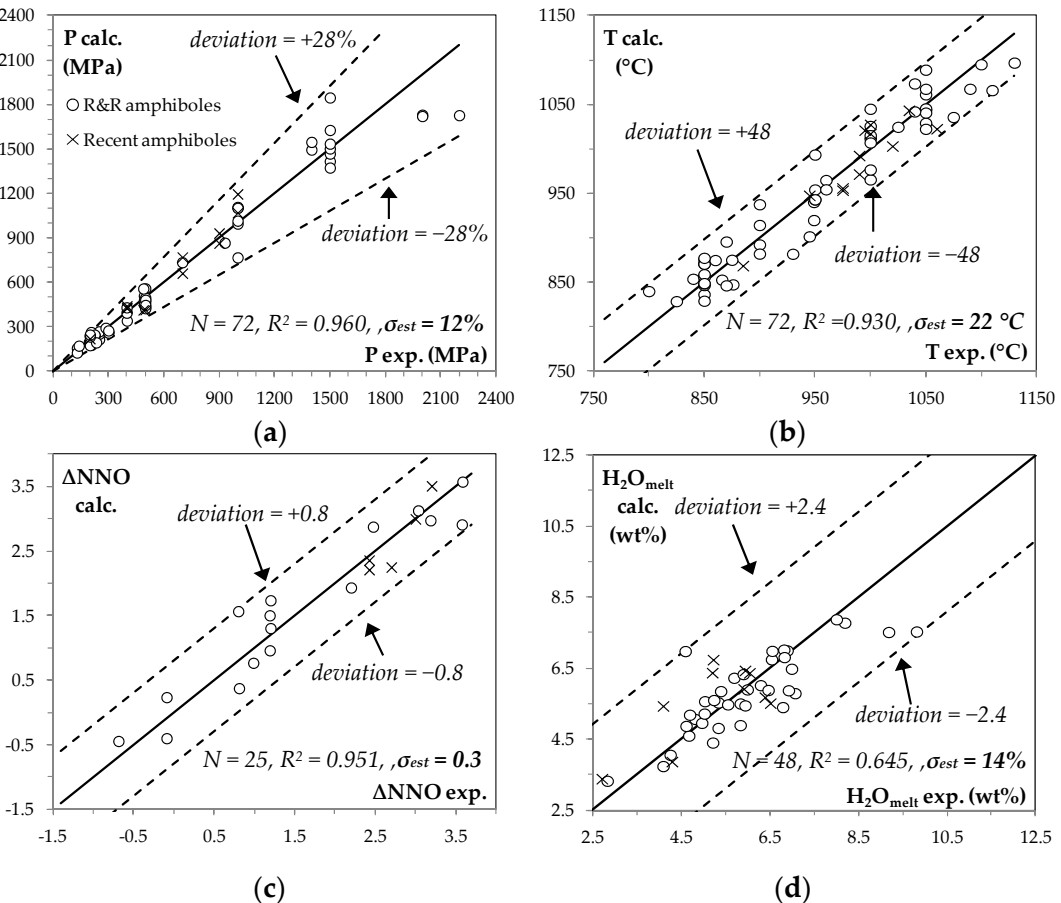

**Figure 1.** Correlations between calculated (calc. Using Amp-TB2.xlsx) and experimental (exp.) physico-chemical conditions ((**a**) P; (**b**) T; (**c**) ΔNNO; (**d**) H$_2$O$_{melt}$). R&R indicates amphibole data selected by [10], whereas recent amphiboles represent the newly selected data in this work (see text and Amp-TB2-xlsx). Overall statistic parameters such as sample size (N; i.e., the overall number of data), determination coefficient (R$^2$), and standard error of the estimate (σ$_{est}$) are also reported (note that the % value is σ$_{est}$ calculated on the percentage error). The 1:1 line (solid) is reported in all diagrams. Dashed lines indicate maximum and minimum deviations (errors) shown and expected by the model.

As an additional check, Figure 2 (and Amp-TB2.xlsx) reports the calibration amphiboles in diagrams such as P-T (Figure 2a), logfO$_2$-T (Figure 2b), and T-H$_2$O$_{melt}$ (Figure 2c). The physico-chemical conditions estimated with Amp-TB2.xlsx using the composition of natural amphiboles should fall within the blue curves reported in all diagrams. Outside these curves, the uncertainties of the thermobarometric method are unknown. In Figure 2a the blue curves most probably indicate the stability of Mg-rich calcic amphibole crystallizing at magmatic steady-state (equilibrium) conditions since the experimental amphiboles cover most of the compositional and physico-chemical condition fields of volcanic ones [10,13]. To some extent, this is also valid for the field included in Figure 2b as the related experimental amphiboles were synthesized at conditions up to 1500 MPa (Amp-TB2.xlsx). In contrast, the blue curves in Figure 2c only delimit the application field of the method since they include a much smaller sample of experimental amphiboles (25; Figure 1c synthesized in narrower condition ranges i.e., 800–1040 °C; 200–700 MPa; −2.1 < ΔNNO < 3.6). In practice, Equation (1) is mostly valid for magmatic amphiboles crystallizing in the crust and upper mantle since its behavior and uncertainty at shallow (<200 MPa) and deep mantle conditions (>700 MPa) are not predictable.

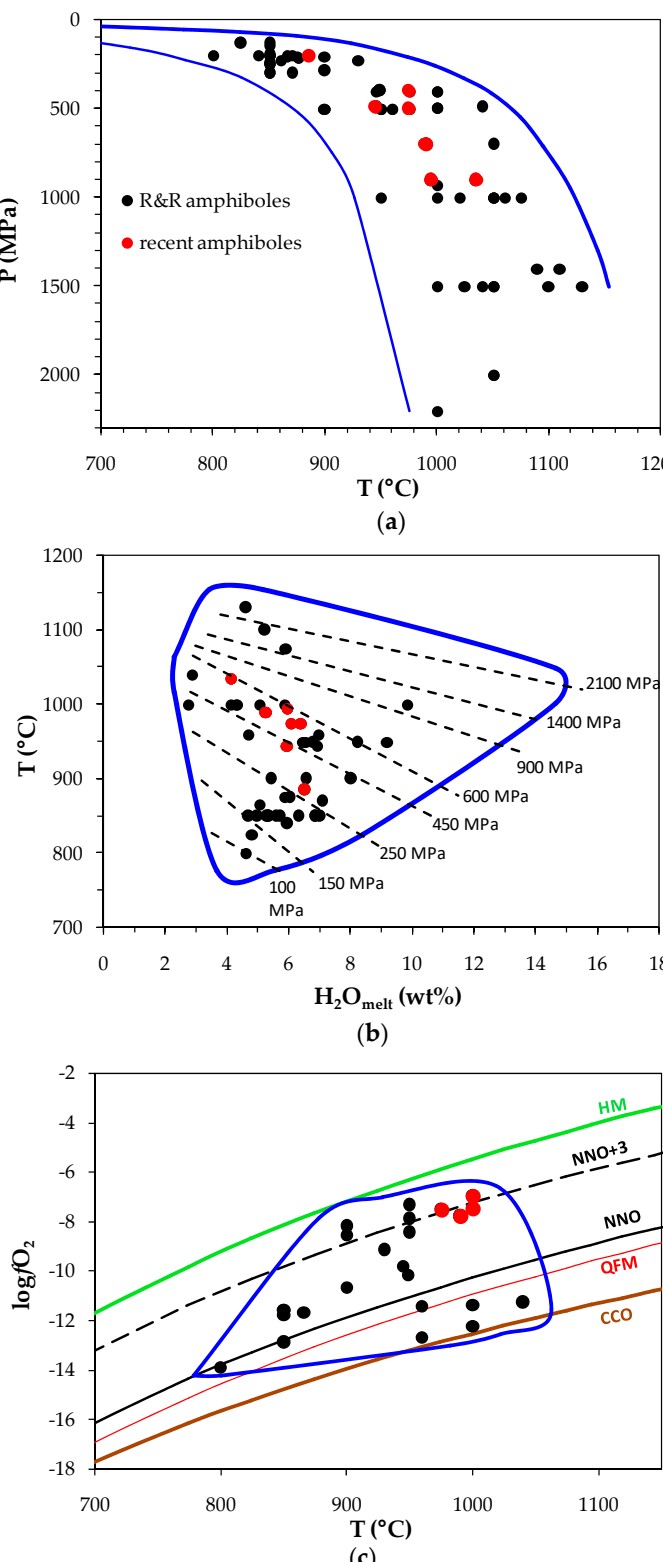

**Figure 2.** (**a**) P vs. T, (**b**) T vs. $H_2O_{melt}$, and (**c**) $\log fO_2$ vs. T diagrams for the experimental amphiboles selected in this work (both recent and those in [10], i.e., R&R). The blue curves are drawn by considering the uncertainties of Amp-TB2.xlsx and delimit the validity of the method. The dashed lines in (**b**) show isobars, whereas (**c**) reports the most common metal-metal oxide buffers according to [45].

The diagrams in Figure 2 are commonly used in petrology and volcanology to display changes in physico-chemical conditions in the magma feeding systems [13,20,25,26,46]. For this reason, Amp-TB2.xlsx includes additional P-T, T-H$_2$O$_{melt}$, and logfO$_2$-T diagrams which can be easily used to input amphibole thermobarometric results. In addition, the T-H$_2$O$_{melt}$ diagram (Figure 2b) includes isobars calculated according to Amp-Tb2.xlsx, which can be used to countercheck the results.

For the application of this thermobarometric model, it is highly recommended to check the quality of the natural Amp compositions using AMFORM.xlsx [40] and follow the approach suggested by [25,26]. To get tight, physico-chemical constraints, it is suggested to avoid the application of Amp-TB2.xlsx to amphiboles crystallized hydrothermally (i.e., from a gaseous phase; e.g., [13]) and to volcanic amphiboles showing disequilibrium textures (e.g., microlites, phenocrysts with acicular, elongate, and hopper morphologies). It is also highly recommended to perform rim-rim or core-rim transect EMP analyses for identifying intra-crystal domains with a homogeneous composition and evaluate (and display into the diagrams of Figure 2) the average and standard deviation of the calculated physico-chemical parameters for each crystal [25].

The results of this single-Amp application to recently erupted volcanic products should be in agreement with constraints from complementary methodologies such as magma storage depths estimates by geophysical data [13,19,26,30–37].

Finally, it is worth noting that testing Amp-TB2.xlsx with experimental amphiboles may lead to misleading results [26,38]. To obtain a homogeneous density of data within the calibration physico-chemical ranges of the five P equations (see above), several literature experimental amphiboles were, discarded although they show relatively high-quality [10]. This means that the application of Amp-TB2.xlsx to additional experimental amphibole data with moderately high-quality will show P, T, *f*O$_2$, and H$_2$O$_{melt}$ values consistent with the experimental ones. In contrast, the application of Amp-thermobarometry to inhomogeneous experimental amphiboles and/or showing disequilibrium textures will apparently (and erroneously) result in physico-chemical large deviations since the chemical equilibrium was not sufficiently approached during the experiments [26].

**Supplementary Materials:** The following are available online at https://www.mdpi.com/2075-163X/11/3/324/s1, Amp-TB2.xlsx. References [18,27–29,41–44,47–66] are cited in the Supplementary Materials.

**Funding:** This research was funded by the Alexander von Humboldt-Stiftung, with a research fellowship to the author, and the Deutsche Forschungsgemeinschaft (DFG), grant number 60422335.A.

**Data Availability Statement:** Data supporting reported results can be found in the Supplementary Materials Amp-TB2.xlsx.

**Acknowledgments:** F. Holtz and many other scientists at the Institut für Mineralogie at LUH are thanked for their support.

**Conflicts of Interest:** The authors declare no conflict of interest.

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
