# Peer review of "Amp-TB2: An Updated Model for Calcic Amphibole Thermobarometry"

_minerals, doi:10.3390/min11030324_

Round 1

Reviewer 1 Report

The author presents an updated model for reconstructing the magmatic pressure, temperature, oxygen fugacity, and water content conditions during amphibole crystallisation. The new model is based on previous work by the author and other colleagues, and incorporates relevant experimental data that has been published recently. Addition of the new data permits lower error estimates on the calculated magmatic conditions, and expands the calibration range of the model.

In the manuscript, the author has concisely described the updates to the new thermobarometric model and given appropriate background commentary on the development of the initial model. It is generally easy to follow and will be useful to any petrologists interested in using calc-alkaline amphibole to trace equilibrium crystallisation conditions in the shallow-deep crust, and where coexisting liquid compositions cannot be constrained.

The accompanying spreadsheet is also clearly structured, and easy to input/output data from. 

My only major comment would be that a brief discussion of the effect of halogen (F, Cl) incorporation on major element stoichiometry may be useful here. Previous experimental work (e.g. Iveson et al., 2017; https://doi.org/10.1093/petrology/egy011) has shown that the Mg–Cl and Fe-F crystallographic avoidance principles in amphibole will affect subsequent calculations of P, T, fO2, and melt H2O contents, due to associated changes in major element compositions. As F- and Cl-rich amphibole are not included in the new experimental dataset used to calibrate the updated model, it could be pertinent to mention these limitations. Indeed, inputting the F-rich amphibole compositions using the attached spreadsheet from the Iveson et al. (2017) experiments shows consistent overestimation of the fO2  relative to the known experimental conditions. However, encouragingly, experimental T, P, and melt H2O contents are still generally reproduced quite closely, and mostly within the errors reported in the manuscript abstract.

I have three other very minor comments, as follows:

Line 167: Should be ‘In practice’

The graphs/axes in Fig. 1 appear to have two black outlines and two grey outlines? Is this intentional?

Should a 'Conclusions' section also be included, despite the brevity of the manuscript?

Other than these relatively minor comments, I do not feel that the manuscript requires significant revision, and should be accepted for publication. It will be widely applicable and utilised.

Reviewer 3 Report

Please see attachment below.

Reviewer 4 Report

This paper provides an updated amphibole single-phase thermobarometer, on the basis of the author’s previous work of Ridolfi and Renzulli (2012), with a slightly extended dataset (with added 12 amphibole compositions). This paper is clearly written, and the method and results seem nice. It deserves publication in Minerals, and a wide application can be expected.
However, I believe that, before acceptance, the following issues should be addressed carefully by the author.
(1) For a robust method, independent tests using a dataset (experimental or natural samples) are a must to show its reliability and real uncertainty in potential applications. Although this manuscript is noted as Short Review or Communication, the missing part may lead readers to question the robustness of the proposed thermobarometer.
(2) In the attached Amp-TB2, amphibole formular is also calculated from input composition, even with a species name. However, the author failed to note in this paper that, this calculation approach is incomplete, because O2- at W site is artificially omitted. It is not consistent with the most recent recommended nomenclature (Hawthorne et al. 2012). Related to this, Fe3+/Fe2+ may be wrongly estimated, which would further affect estimations for ΔNNO and H2O content.
(3) Physical rationality of the estimation equations is expected from the author, i.e., are these equation has some physical meaning or implications?
(4) Line 116: How is the “overall pressure uncertainty of ±12%” obtain? Mathematical meaning should be noted explicitly. It is the same question for all the other estimations plotted in Fig. 1. Is this calculated σ represents a reliable estimation for the uncertainty of an input natural amphibole?

Round 2

Reviewer 4 Report

I am satisfied with the replies and revisions made by the author, and therefore recommend acceptance of this paper by Minerals.